# Structural Component Identification and Damage Localization of Civil Infrastructure Using Semantic Segmentation

**DOI:** 10.3390/s25154698

**Published:** 2025-07-30

**Authors:** Piotr Tauzowski, Mariusz Ostrowski, Dominik Bogucki, Piotr Jarosik, Bartłomiej Błachowski

**Affiliations:** 1Institute of Fundamental Technological Research, Polish Academy of Sciences, 02-106 Warsaw, Poland; ptauzow@ippt.pan.pl (P.T.); mostr@ippt.pan.pl (M.O.); dbogucki@ippt.pan.pl (D.B.); pjarosik@ippt.pan.pl (P.J.); 2IDEAS NCBR Sp. z o. o., 00-801 Warszawa, Poland

**Keywords:** semantic segmentation, structural health monitoring, computer vision-based techniques

## Abstract

Visual inspection of civil infrastructure for structural health assessment, as performed by structural engineers, is expensive and time-consuming. Therefore, automating this process is highly attractive, which has received significant attention in recent years. With the increasing capabilities of computers, deep neural networks have become a standard tool and can be used for structural health inspections. A key challenge, however, is the availability of reliable datasets. In this work, the U-net and DeepLab v3+ convolutional neural networks are trained on a synthetic Tokaido dataset. This dataset comprises images representative of data acquired by unmanned aerial vehicle (UAV) imagery and corresponding ground truth data. The data includes semantic segmentation masks for both categorizing structural elements (slabs, beams, and columns) and assessing structural damage (concrete spalling or exposed rebars). Data augmentation, including both image quality degradation (e.g., brightness modification, added noise) and image transformations (e.g., image flipping), is applied to the synthetic dataset. The selected neural network architectures achieve excellent performance, reaching values of 97% for accuracy and 87% for Mean Intersection over Union (mIoU) on the validation data. It also demonstrates promising results in the semantic segmentation of real-world structures captured in photographs, despite being trained solely on synthetic data. Additionally, based on the obtained results of semantic segmentation, it can be concluded that DeepLabV3+ outperforms U-net in structural component identification. However, this is not the case in the damage identification task.

## 1. Introduction

Maintaining the traffic infrastructure in good technical condition is a key element of rail and road traffic safety. The expansion of traffic infrastructure increases the problems related to the maintenance of facilities. Therefore, there is a need for intensive monitoring of the technical condition of a large number of objects. The intensive development of vision techniques, together with artificial intelligence, allows for effective automation of the inspection process of traffic infrastructure facilities. Installed cameras, as well as remotely controlled or autonomous vehicles or drones, can independently monitor by capturing images. The next step is image processing by artificial intelligence systems based on deep learning, which recognizes whether a structure has damage or not. To perform this task, it is necessary to train deep neural networks capable of recognizing damage in the images. It is also essential to distinguish parts of the structure in the images to properly locate the damage, as well as to find out how important the structural elements that occur are. Convolutional neural networks seem to be one of the most suitable tools for this demanding task.

The fundamentals of neural network architecture date back to the 1960s. Hubel et al. [1] proposed the concept of receptive fields based on their study of the visual cortex in monkeys. Their findings influenced the idea of local receptive fields in convolutional neural networks (CNNs), where each convolutional filter detects localized features within small regions of an image. The hierarchical feature detection model of the visual cortex further inspired the multi-layer architecture of CNNs, with early layers capturing simple features and deeper layers identifying more complex patterns.

The biological concept of shared weights in the visual cortex led to the implementation of weight sharing in CNN filters, which enables translation equivariance in feature detection. In 1974, Werbos [2] introduced the concept of gradient-based learning for neural networks, which was mathematically similar to the modern backpropagation algorithm, though it was not yet fully formalized under that name. He referred to it using terms like *gradient descent* and *error correction* for training neural networks. Werbos outlined the fundamental principles of propagating errors through layers to adjust weights. Around the same period, Fukushima [3] developed Neocognitron, a neural network with multiple layers of neurons designed to process increasingly complex visual features. The Neocognitron employed multiple stages to extract features from input images, mirroring the hierarchical processing of the human brain. This model became one of the key precursors to modern CNNs. In this architecture, convolutional layers similarly utilize local receptive fields (filters) to capture small-scale patterns in input images, enabling the network to learn hierarchical feature representations at various scales.

A major breakthrough came in the 1980s when Rumelhart et al. [4] formalized and popularized the backpropagation algorithm, demonstrating its effectiveness in training multi-layer neural networks. Their work provided a practical method for adjusting weights through error propagation, significantly improving learning efficiency. This development marked a turning point, enabling deeper networks to be trained effectively and fueling the rise of modern deep learning. The practical application of CNNs gained momentum with LeCun et al. [5], who demonstrated the effectiveness of gradient-based learning for document recognition, leading to widespread adoption in optical character recognition and early deep learning applications. The field saw a transformative shift in 2012 with Krizhevsky et al.’s [6] AlexNet, which leveraged deep CNNs and GPU acceleration to achieve a breakthrough in the ImageNet competition, dramatically improving image classification performance. Building on these advancements, Simonyan and Zisserman [7] introduced VGG networks, demonstrating that increasing network depth with small convolutional filters further enhances performance.

The evolution of CNNs also extended to semantic segmentation, with Long et al. [8] pioneering fully convolutional networks (FCNs), which replaced fully connected layers with convolutional ones to enable pixel-wise classification. Ronneberger et al. [9] further refined these ideas with U-Net, a model specifically designed for biomedical image segmentation, featuring an encoder–decoder architecture with skip connections for improved localization accuracy. Subsequent research by Chen et al. [10,11,12] led to the development of the DeepLab architecture, introducing atrous (dilated) convolution and fully connected Conditional Random Fields (CRFs) to enhance segmentation accuracy and capture multi-scale contextual information.

In the domain of civil infrastructure monitoring, Ros et al. [13] introduced the SYNTHIA dataset, which consists of synthetic images designed for urban scene segmentation. While primarily developed for autonomous driving applications, SYNTHIA’s realistic textures, lighting conditions, and diverse structural layouts provide valuable insights into training models to segment complex structural elements in civil engineering contexts. Spencer et al. [14] provided a comprehensive review of computer vision applications in structural inspection, describing how deep learning enables damage detection, crack identification, and deformation analysis in bridges and buildings. Their work outlined the transition from traditional manual inspections to automated, vision-based methods, highlighting challenges such as data scarcity, environmental variations, and model generalization. Another excellent overview of computer vision-based techniques for SHM (CV-SHM) was presented by Azimi et al. [15]. In their paper, they discussed various neural architectures such as AlexNet, VGG16 and VGG19, Resnet50, GoogleNet, ZFNet or CrackNet and various software frameworks for CV-SHM applications such as TensorFlow v2.19, PyTorch, Keras v.3.9, Caffe and Theano.

Bao et al. [16] explored the integration of machine learning with structural health monitoring (SHM), emphasizing the role of data-driven approaches in anomaly detection, predictive maintenance, and long-term structural assessment. Their study highlighted how deep learning can process sensor data, thermal imagery, and high-resolution photographs to track structural degradation over time. Bianchi and Hebdon [17] compiled a benchmark dataset for visual structural inspections, containing labeled images of bridges, viaducts, and other civil infrastructure elements with corresponding damage annotations. Their dataset aids the development of robust, generalizable models for detecting cracks, corrosion, and material degradation in large-scale infrastructure networks.

An application of the U-net neural network architecture for crack identification in concrete structures was proposed by Bhowmick et al. [18]. In order to demonstrate the real-life applicability of the proposed approach, the authors conducted the laboratory experiment on an 8-foot-long beam, which is gradually loaded until it failed in flexure.

Recent work in structural and pavement damage segmentation has focused on enhancing model architectures and employing attention mechanisms for improved crack delineation. For example, Cha et al. (2024) [19] reviewed DL-based SHM methods, including nondestructive testing, UAV vision, digital twins, and physics-informed models, emphasizing opportunities for enhanced reliability and automation. Gao et al. (2023) [20] proposed a Multiattribute Multitask Transformer (MAMT2) framework that jointly addresses classification, localization, and segmentation of structural damages, outperforming CNNs on a large benchmark dataset by leveraging inter-task relationships. Azimi and Yang (2024) [21] introduced the Refined Segment Anything Model (R-SAM), which combines zero-shot segmentation with a refinement module to achieve high-accuracy multiclass segmentation without extensive training, supporting tasks like tracking and 3D reconstruction. Shahin et al. (2024) [22] enhanced concrete crack detection using a hybrid Visual Transformer with image enhancement techniques, achieving near-perfect accuracy and demonstrating a fast, efficient alternative to conventional CNNs aligned with Industry 4.0 goals. Shen et al. [23] incorporated boundary refinement within DeepLabV3+, while Yuan et al. [24] and Hang et al. [25] embedded the ECA-UNet– and CBAM-based attention modules into U-Net backbones, achieving significant gains in IoU and recall. Zhou et al. [26] proposed a residual feature pyramid attention network for efficient crack detection in underground utility tunnels. For synthetic data and domain adaptation, Graybeal et al. [27] introduced a residual sharp U-Net (Rs-Net) with multi-scale feature fusion, demonstrating robustness across diverse pavement crack datasets.

Finally, Cheng et al. [28] proposed a random bridge generator, a framework that synthetically creates bridge structures with varying geometries and defect patterns. This synthetic data serves as a training platform for deep learning models, allowing researchers to develop and validate computer-vision-based structural inspection algorithms in a controlled environment before real-world deployment.

In addition to peer-reviewed studies, several recent preprints have explored the use of synthetic and semi-synthetic data to improve segmentation performance in damage detection tasks. For instance, Parslov et al. [29] introduced a procedural pipeline to simulate and annotate damage on 3D vehicle models, showing that a combination of synthetic and real data improves segmentation quality. Dondi et al. [30] proposed a semi-synthetic approach for crack detection under earthquake conditions, using parametric 3D modeling to generate training data. Nowacka et al. [31] applied a 3D U-Net to segment micro-CT images of fiber-reinforced concrete using semi-synthetic labels, while Jaziri et al. [32] employed a fractal-based crack simulator to pretrain a hybrid segmentation model for improved real-world generalization. Although these works are preprints, they reflect current trends in data-centric model development for fine-grained structural damage analysis.

These recent advances reflect the trend toward combining architectural sophistication with data-centric strategies to address the inherent class imbalance and fine-detail segmentation challenges in structural damage detection.

It is also worth mentioning the recently created tool *Segment anything* [33]. Based on the *Vision Transformer* architecture, it is capable of segmenting any image. However, due to the high requirements of the presented architecture, it is not an alternative in all applications today.

These advancements provide the foundation for applying semantic segmentation techniques to structural viaduct inspection, enabling automated structural part identification and damage localization with a deep learning approach.

The objective of this work is to propose a training procedure for state-of-the-art models for two neural networks: U-net and DeepLabV3+. The effectiveness of the trained models in the semantic segmentation of civil infrastructures has been demonstrated on two tasks: structural component identification and damage localization. Both tasks have been tested on the Tokaido dataset, containing realistic images of high-speed railway viaducts.

The remainder of the paper is organized as follows: In the second section, the methodology for semantic segmentation of civil infrastructure images is proposed. The third section describes the results of the semantic segmentation using two neural networks, namely Unet and DeepLabV3+. A comparison of the results obtained using these two networks indicates that DeepLabV3+ outperforms U-net in both component identification and damage detection tasks. The paper concludes with a summary and enumeration of the most important observations from our research.

## 2. The Proposed Methodology for Semantic Segmentation of Civil Infrastructure Images

### 2.1. Overview of the Proposed Methodology for the Training Process

The proposed methodology for the semantic segmentation of images of civil engineering structures relies on a deep learning approach and a synthetic dataset utilized in the training process. The training process consists of four main steps described below and presented in Figure 1:

*Step 1.* Data Preparation—Image acquisition

The first step of the proposed methodology involves the acquisition of images of real engineering structures. This goal can be achieved, for example, using autonomous cars or unmanned aviation vehicles (UAVs). The image acquisition process itself usually requires a high-resolution camera with image stabilization.

*Step 2.* Preprocessing of collected data—Image augmentation

Neural networks trained solely on original data suffer from a weak robustness property. Therefore, in the second step of our methodology, we apply image augmentation. This augmentation involves introducing small perturbations to the training set, expressing imperfections in data acquisition, such as contrast, brightness, exposure, or matrix noise.

*Step 3.* Training process—Selection of neural network architecture

The quality of the results obtained using deep learning strongly depends on the neural network architectures used. Therefore, in the third step, two different architectures were proposed, namely, the U-net and DeepLabV3+. These are models containing tens of millions of parameters, and their training on modern computers equipped with graphics cards takes from a few to a dozen or so hours.

*Step 4.* Making predictions—Semantic segmentation

The final step of the proposed methodology is related to making predictions for both semantic segmentation tasks, i.e., structural component identification and damage detection. The first task provides information about the probability of assignment of the individual pixels in the image to a selected category of structural components such as beams or columns. The second task reveals areas of the image with concrete spalling or reinforcement exposure.

### 2.2. Data Set Description

To train a neural network model, a substantial set of training and validation data is needed. This task uses a dataset of 7575 artificially generated images of viaducts, created using Blender3D, a computer graphics system, based on 3D models created by [34]. The dataset also includes a depth channel and eight segmentation mask channels. The number of masks corresponds to the number of structural object types being detected in the images and indicates their locations. This dataset forms the basis for neural network training.

The images illustrate a 3D model of a straight section of a railway viaduct in two versions: one with crossbeams between the columns and one without. The structural elements are textured with images, most commonly concrete, though stone textures are also used. The ground is textured with various images, ranging from wood to stubble, and occasionally with unusual textures that are difficult to identify.

There are approximately 100 different versions of this scene, each featuring different textures. For each version, several dozen images of the viaduct are captured, typically along its axis. Interestingly, there are no views taken perpendicular to the viaduct in the dataset.

At a certain distance from the viaduct (approximately equal to its height), vertical panels parallel to the viaduct are placed. These panels display various urban, landscape, or even graffiti images to simulate different real-world environments surrounding the viaduct. Below are some examples of the different texture variations (see Figure 2).

In addition to the visual inspection of the data, which has a qualitative nature, certain quantitative characteristics should be established. For the multi-class semantic segmentation problem, one of the basic values is the empirical distribution of pixels belonging to individual classes.

The pie chart illustrating the pixel distribution for the presented viaduct images is depicted in  Figure 3. The chart shows that the most pixels belong to the non-structural class. This is explained by the fact that most shots of the viaduct are taken from middle distance and include the background, sky, or ground. Images of details of the structure are in the minority. As for structural elements such as beams, slabs, or columns, there is a slight predominance of columns. The reason for this quantitative advantage is the fact that the columns are a more exposed part of the structure than the slabs or beams and therefore take up more pixels in the photos. The least pixel-occupying objects are non-structural elements such as poles, cables, fences, tracks, or railway sleepers. These elements occupy small areas of the image in comparison with structural elements. Furthermore, there are not many images of the track in the dataset.

### 2.3. Training Protocol and Learning Rate Schedule

All models were trained using the Adam optimizer with an initial learning rate of $1\times 10^{-4}$. To ensure efficient convergence and prevent overfitting, we employed three training callbacks:**ModelCheckpoint:** During training, the model was monitored using the validation loss. The best-performing model (with the lowest validation loss) was saved to disk. This ensured that the final model used for evaluation did not suffer from overfitting or suboptimal convergence due to later epochs.**ReduceLROnPlateau:** To dynamically adapt the learning rate during training, we used a reduction-on-plateau strategy. If the validation loss did not improve for 5 consecutive epochs, the learning rate was reduced by a factor of 0.5. This allowed the optimizer to take smaller steps during later training stages, which helped stabilize convergence and fine-tune the weights.**EarlyStopping:** Training was halted if the validation loss did not improve for 10 consecutive epochs. The weights from the best epoch (according to validation loss) were automatically restored, ensuring that the final model did not overfit.

This training protocol led to smooth convergence across experiments. The learning rate typically underwent 1–3 reductions per training session, resulting in final learning rates between $1\times 10^{-4}$ and $2.5\times 10^{-5}$, depending on the model configuration. The combination of adaptive learning and early stopping contributed to both training stability and generalization to the test set.

Recently, the *Segment Anything* model was developed by Kirillov et al. [33]. This neural network, based on the *Vision Transformer* architecture, performs semantic segmentation of any image. We tested this promising solution on our photos. However, it quickly became apparent that it would not be a suitable alternative to our solution. First, the model is ten times larger, requiring a powerful graphics card for predictions. In contrast, our model can be loaded onto a website, and predictions can be easily performed by a standard processor. Second, the model’s precision is significantly lower. As Figure 4 demonstrates, it fails to distinguish between a slab and a beam. Finely training such a large model is a process that exceeds the capabilities of even a high-end desktop computer. Third, the segmentation results in a multitude of classes, sometimes even within the same category (e.g., viaduct columns), where individual columns are assigned different classes. Consequently, the classification results necessitate additional post-processing. Therefore, we conclude that a smaller, dedicated model, specifically trained for our purposes, would be far more effective.

### 2.4. Image Augmentation

In the task of semantic segmentation for engineering structures, data augmentation plays a crucial role in enhancing model robustness and generalization. The image set presented in Figure 5 illustrates several augmentation techniques applied to viaduct scenes, ensuring better performance in diverse conditions. The brightness, contrast, and gamma adjustments modify the illumination of the image, simulating varying lighting conditions, which is essential for real-world applications where light exposure fluctuates. Usually, the above concepts are intuitively understandable; however, for the sake of scientific accuracy, we will provide below the mathematical formulas for the above image transformations. Let $I_{\mathrm{in}}\left(x,y\right)$ denote the input image intensity at pixel location (x,y), and $I_{\mathrm{out}}\left(x,y\right)$ the output intensity after transformation. All intensity values are assumed to be normalized to the interval [0,1].

(a)*Original image.* In formal mathematical language, the usage of the original image is equivalent to identity mapping between the input and output images:$$I_{\mathrm{out}}\left(x,y\right)=I_{\mathrm{in}}\left(x,y\right)$$(b)*Brightness Adjustment.* A brightness shift adds a scalar bias β to each pixel:$$I_{\mathrm{out}}\left(x,y\right)=\mathrm{clip}\left(I_{\mathrm{in}}\left(x,y\right)+\beta ,0,1\right).$$Here, clip(a,0,1) limits the value to the range [0,1].(c)*Contrast Adjustment.* Contrast is adjusted by scaling the deviation from the mean intensity μ:$$I_{\mathrm{out}}\left(x,y\right)=\mathrm{clip}\left(\alpha \;[I_{\mathrm{in}}\left(x,y\right)-\mu ]+\mu ,0,1\right),$$
where α is the contrast factor and$$\mu =\frac{1}{WH}\sum_{x=1}^{W}\sum_{y=1}^{H}I_{\mathrm{in}}\left(x,y\right)$$
is the mean image intensity over a W×H image.(d)*Gamma Correction.* Gamma correction applies a nonlinear transformation:$$I_{\mathrm{out}}\left(x,y\right)=I_{\mathrm{in}}{\left(x,y\right)}^{\gamma }$$The parameter γ controls the shape of the correction curve, where for γ<1, the image appears brighter, while for γ>1, the image appears darker.(e)*Noise injection.* Noise injection introduces random perturbations, making the model more resilient to sensor noise, especially in shadows. Gaussian (normal) noise was used for each image in the input batch.$$I_{\mathrm{out}}\left(x,y\right)=\mathrm{clip}\left(I_{\mathrm{in}}\left(x,y\right)+N\left(0,\sigma \right),0,1\right)$$Augmentation function generates pixel-wise noise drawn from a Gaussian distribution with a mean value 0 and variable standard deviation given in Table 1.(f)*Flipping.* Flipping generates different perspectives of the viaduct, which helps the model learn orientation-independent features. In our application, horizontal flipping was used$$I_{\mathrm{out}}\left(x,y\right)=I_{\mathrm{in}}\left(W-x,y\right)$$(g)*Rotation.* Rotation transformation R(θ) can be described as follows:$$I_{\mathrm{out}}\left(x,y\right)=I_{\mathrm{in}}\left(x^{\prime },y^{\prime }\right),$$
where$$\left[\begin{array}{c} x^{\prime }\\ y^{\prime } \end{array}\right]=\mathrm{clip}\left(R\left(\theta \right)\left[\begin{array}{c} x\\ y \end{array}\right],\left[\begin{array}{c} 0\\ 0 \end{array}\right],\left[\begin{array}{c} W\\ H \end{array}\right]\right),$$
and θ is the rotation angle.(h)*CutMix technique.* CutMix, a technique that replaces a region of an image with an image taken from another sample (see Figure 5h). It helps in learning more discriminative features by forcing the model to focus on multiple context regions. In the CutMix augmentation strategy, a rectangular region from one training image $x_{a}$ is copied and pasted into a target image $x_{b}$. The pasted region is defined by its top-left corner coordinates $\left(x_{0},y_{0}\right)$ and size (w,h). The target image and corresponding label are modified as follows:(1)$$x^{\prime }=M⊙x_{a}+\left(1-M\right)⊙x_{b},\;y^{\prime }=M⊙y_{a}+\left(1-M\right)⊙y_{b},$$
where $M\in {\left\{0,1\right\}}^{H\times W}$ is a binary mask that equals 1 within the rectangular region and 0 elsewhere, and ⊙ denotes element-wise multiplication. The values of $x_{0}$, $y_{0}$, *w*, and *h* are sampled uniformly within valid image bounds to ensure that the pasted region fits entirely within the image dimensions.

Although CutMix was included as one of the augmentation strategies in this study, we note that its effectiveness for damage detection tasks may be limited. The sharp rectangular boundaries introduced by standard CutMix are visually artificial and may be easily identified by the network, which could reduce the augmentation’s intended effect. In future work, we plan to explore more realistic and adaptive data mixing techniques, such as irregular mask-based blending or texture synthesis methods, which better approximate the spatial characteristics of actual damage.

These augmentations, combined with the original images, create a richer dataset, ultimately improving the segmentation accuracy and robustness in real-world deployments. The variability values of individual augmentation parameters that were used in the implementation are presented in Table 1.
sensors-25-04698-t001_Table 1Table 1Augmentation parameters with probability of occurrence.AugmentationParametersProbability [%]Brightnessβ=[−0.2,0.2]40Contrastα=[0.8,1.2]40Gammaγ=[0.7,1.3]40Noiseσ=[0.01,0.05]40Rotationθ=[−18∘,+18∘]60FlipN/A50CutMixposition (x0,y0), size (w,h)40

The probabilities assigned to each augmentation technique were selected based on a combination of insights from the existing literature, empirical experimentation, and the specific characteristics of the Tokaido synthetic dataset used in this study.

**Brightness, Contrast, Gamma, and Noise (40%)**: These augmentations simulate varying image acquisition conditions, such as lighting inconsistencies and sensor noise, which are common in UAV-based inspections. A probability of 40% offers a balanced trade-off—frequent enough to improve model generalization but not so dominant as to degrade data fidelity. This rate is consistent with augmentation strategies adopted in related deep learning studies on civil infrastructure monitoring [14,15].**Flip (50%)**: Horizontal flipping is applied with a probability of 50%, a standard setting in many vision-based learning pipelines. This is particularly relevant for viaduct imagery, which often exhibits axial symmetry. A 50% rate introduces orientation variation while preserving the structural coherence of the scene.**Rotation (60%)**: A slightly higher probability was chosen for rotation to reflect the diversity of camera angles typically encountered in UAV inspections. The selected range of $\theta \in [-18^{\circ },+18^{\circ }]$ corresponds to realistic off-axis views without introducing distortions. A 60% probability ensures adequate rotational diversity, which was empirically shown to enhance performance in both segmentation tasks.**CutMix (40%)**: CutMix introduces strong contextual perturbations by blending regions from different images. While this encourages the model to learn more robust features, excessive use can reduce semantic coherence, particularly in structured scenes like viaducts. Hence, a conservative value of 40% was adopted.

The chosen probabilities were also verified through validation experiments. Increasing them further (e.g., above 70%) often led to decreased performance or instability during training, especially for the damage detection task, where fine details are critical. Thus, the selected values reflect a carefully tuned balance between diversity and realism in the augmented data.

### 2.5. Architectures of Neural Networks for the Semantic Segmentation Problem

The problem of multi-class semantic segmentation was solved using deep learning methods. The most effective neural architectures in image processing are convolutional neural networks because they can learn to recognize certain patterns regardless of their location in the image (equivariance and invariance to feature translation). In this study, U-net and DeeplabV3+ neural networks are employed. Diagrams of these architectures are presented in Figure 6 and Figure 7. These models are implemented in the Keras 3.9, Python 3.12.9 package with the Tensorflow 2.19 backend.

#### 2.5.1. U-Net Architecture with Optional Attention and Customizations

As a first architecture of the neural network in this study, we employ a customizable U-Net (Figure 6). Our implementation is based on Keras API, which provides support for several advanced features such as attention gates, flexible upsampling, and dropout control. The model is implemented in a modular fashion to allow adaptation for different semantic segmentation tasks:*Input and output*. The network accepts input images of arbitrary spatial dimensions, specified by the input_shape parameter, and produces a segmentation mask with either a single channel (sigmoid activation for binary segmentation) or multiple channels (softmax for multi-class segmentation), controlled by the num_classes and output_activation arguments.*Encoder (Contracting Path)*. The encoder consists of a configurable number of levels (num_layers). At each level:-A double convolution block is applied.-Each convolution block may include optional batch normalization, spatial or standard dropout, and ReLU activation (or other activation functions).-After the convolution block, max pooling is applied to downsample the feature maps by a factor of 2.-The number of filters starts at a base value (filters, typically 16) and doubles at each subsequent level.-The dropout rate can increase with each layer to gradually regularize deeper layers.*Bottleneck*. After the encoder, a central convolution block (the bottleneck) captures high-level features. This block uses the highest number of filters and the final dropout level before upsampling begins.*Decoder (Expanding Path)*. The decoder path reconstructs the segmentation mask through upsampling: Each upsampling step uses either transposed convolution (Conv2DTranspose) or nearest-neighbor upsampling followed by convolution, as selected by upsample_mode (deconv or simple). The feature maps from the corresponding encoder level are concatenated via skip connections to retain spatial details. Optionally, attention gates can be applied to modulate skip connections. These gates compute attention coefficients based on both the encoder and decoder features, enhancing relevant spatial regions and suppressing irrelevant ones. A convolution block follows each concatenation to refine the fused features.*Output layer* Final 1 × 1 convolution reduces the number of output channels to num_classes, and the activation function (sigmoid or softmax) produces the pixel-wise class probabilities.*Attention Mechanism*. The optional attention gate mechanism follows the additive attention formulation. It uses 1 × 1 convolutions on both the decoder input and the encoder skip connection to compute intermediate features, which are added and passed through ReLU and sigmoid activations to produce an attention mask. This mask is then applied multiplicatively to the skip connection before concatenation, effectively guiding the model to focus on relevant spatial regions.

#### 2.5.2. DeepLabV3+ Network Architecture

We also employed a customized implementation of DeepLabV3+ (Figure 7) for semantic segmentation, constructed using TensorFlow and Keras. The model follows the standard DeepLabV3+ architecture, integrating a dilated convolution-based encoder and a multi-scale context module, followed by a decoder for fine spatial recovery.

The key components of the DeepLabV3 + architecture are described below:*Backbone Feature Extractor.* The encoder utilizes a ResNet101V2 backbone pretrained on ImageNet and excludes the top classification layers. Intermediate feature maps are extracted from-conv4_block6_2_relu (deep feature map) for the context module;-conv2_block3_2_relu (early feature map) for spatial detail recovery in the decoder.*Atrous Spatial Pyramid Pooling (ASPP).* We adopt a modified ASPP block that applies convolutions with different dilation rates to capture multi-scale contextual information. Specifically, this module includes global average pooling followed by a 1×1 convolution and upsampling; parallel 1×1 and 3×3 convolutions with dilation rates of 4, 6, 12, and 18; concatenation of all branches; and a final 1×1 convolution to aggregate features. An extended variant DilatedSpatialPyramidPoolingD4 is also defined and tested, supporting finer granularity via additional dilation rates (e.g., 4, 8, …, 24), but it is not used in the main model function due to the minor impact of the 24 dilation rate observed.*Decoder and Upsampling.* To refine segmentation boundaries, the decoder combines the ASPP output with high-resolution features from the early encoder stage. The decoder consists of a 2×2 transposed convolution applied to the ASPP output (upsampling by a factor of 2), a 1×1 convolution applied to the early feature map for dimension alignment, the concatenation of both feature streams, two convolutional refinement blocks, and further upsampling using transposed convolutions (with strides of 4 and 2) and interleaved with ReLU-activated convolutions. The final prediction layer is a 1×1 convolution with num_classes output channels and the specified activation function (softmax or other).

## 3. Loss Function and Evaluation Metrics

A loss function is a critical component in deep learning that quantifies the difference between a model’s predictions and the actual target values. It essentially measures how well a model is performing and guides the optimization process by minimizing the loss, leading to improved accuracy. In this study, **categorical cross-entropy** is used as the loss function, which can be expressed as follows:$$L=-\sum_{i=1}^{N}y_{i}\log\left(p_{i}\right),$$
where *N* is the number of categories, $y_{i}$ is the true probability distribution (one-hot encoded vector), and $p_{i}$ is the predicted probability distribution. While effective for general classification, categorical cross-entropy proved suboptimal for fine-grained damage segmentation due to extreme class imbalance, particularly when the target damage regions (e.g., cracks and reinforcement) occupy a small fraction of the image.

To address this, we adopted the **weighted focal Tversky loss**, which is more suitable for semantic segmentation tasks with rare classes. The Tversky index generalizes the Dice coefficient by introducing asymmetric weighting of false positives and false negatives:$${\mathrm{Tversky}}_{c}=\frac{{\mathrm{TP}}_{c}+\epsilon }{{\mathrm{TP}}_{c}+\alpha \cdot {\mathrm{FP}}_{c}+\beta \cdot {\mathrm{FN}}_{c}+\epsilon },$$
where ${\mathrm{TP}}_{c}$, ${\mathrm{FP}}_{c}$, and ${\mathrm{FN}}_{c}$ represent the number of true positive, false positive, and false negative pixels for class *c*, respectively. After hyperparameter tuning, we selected α=0.3, β=0.7 to prioritize recall—a key objective in damage detection. The Tversky index was extended into a focal form to concentrate learning on hard-to-classify regions:$$L_{\mathrm{FT}}=\sum_{c=1}^{C}w_{c}\cdot {\left(1-{\mathrm{Tversky}}_{c}\right)}^{\gamma },$$
with γ=1.5 and class weights $w_{c}=\left[0.05,0.3,0.65\right]$ assigned to background, cracks, and reinforcement, respectively. This weighting scheme downplayed the dominant background class and emphasized minority damage classes.

To quantify the results obtained by the two above network architectures, one has to select the appropriate evaluation metrics. In this study the following metrics have been applied:

Accuracy is a metric that tells us how close a given set of classified pixels is to their ground truth$$\mathrm{Accuracy}=\frac{N_{c}}{N_{t}},$$
where $N_{c}$ represents the number of correct predictions and $N_{t}$ is related to the total number of predictions.

It is important to emphasize that the accuracy metric is not particularly suitable for our application, especially in the context of damage detection. Accuracy is computed as the ratio of correctly classified pixels (including true positives and true negatives) to the total number of pixels. In the semantic segmentation of structural damage, the vast majority of pixels belong to the background class (true negatives), while damage regions such as cracks or spalling represent only a small fraction of the image. Consequently, a model that completely fails to localize the damage may still achieve deceptively high accuracy (e.g., >90%) simply by correctly classifying the background. This renders the metric insensitive to errors in the small, yet critical, damage regions. For this reason, we rely on evaluation metrics such as Mean Intersection over Union (mIoU), Recall, Precision, and F1 score (Dice coefficient), which are more robust to class imbalance and better reflect performance on the underrepresented damage classes.

The Mean Intersection over Union (mIoU) metric quantifies the overlap between the predicted and ground truth segmentation masks by computing the ratio of the intersection area (true positives) to the union of the two masks, which includes both false positives (FP) and false negatives (FN) (Figure 8). This metric treats over-segmentation (FP) and under-segmentation (FN) symmetrically. In the semantic segmentation of structural elements, mIoU is well suited, as the objective is to delineate the full extent of structural components with high spatial accuracy.

However, in the context of damage detection, the relative importance of false positives and false negatives is not symmetric. False positives—i.e., overestimations where non-damage pixels are misclassified as damage—are generally less critical than false negatives, where actual damage is missed. In safety-critical applications, missing a damaged region can have severe consequences. Therefore, Recall and Precision metrics offer more informative evaluation in this setting. Both relate the area of correctly predicted damage (true positives, TPs) to only one type of error:(2)$$\mathrm{Precision}=\frac{TP}{TP+FP}$$(3)$$\mathrm{Recall}=\frac{TP}{TP+FN}$$

To assess the trade-off between these two error types, we also report the F1 score, defined as the harmonic mean of Precision and Recall:(4)$$F1\;\mathrm{Score}=2\cdot \frac{\mathrm{Precision}\cdot \mathrm{Recall}}{\mathrm{Precision}+\mathrm{Recall}}$$

The F1 score provides a single summary metric that balances the two and is especially useful when the importance of avoiding false negatives and false positives needs to be jointly considered. In our case, while Recall remains the most critical metric due to the risk of missing damage, the F1 score allows us to monitor overall detection quality in a more balanced way.

For reference, the mIoU metric is defined as(5)$$\mathrm{mIoU}=\frac{TP}{TP+FP+FN}$$
where TP, FP, and FN refer to the number of true positives, false positives, and false negatives, respectively, computed per class.

This comprehensive metric suite provides a more damage-sensitive evaluation. Notably, the reinforcement class, previously undetected under cross-entropy loss, achieved an F1 score of 0.61 and IoU of 0.44 with the proposed loss function, while crack detection improved to an F1 of 0.65 and an IoU of 0.48—demonstrating the effectiveness of the adapted training strategy.

After defining the loss function and metrics used for the training process, one can characterize the dimensions of the input and output layers of the proposed neural network architecture. Figure 9 illustrates the process for the structural component identification task. It depicts a deep learning pipeline utilizing DL models such as U-Net and DeepLabV3+.

The input to the models is shown as 320 × 160 × 3 data. The last dimension in the input represents the three color channels: red (R), green (G), and blue (B).The output of the models for this task is represented as 320 × 160 × 4 with four classes.The Figure 9 also indicates that the segmentation process distinguishes between different structural components and the background, assigning pixels to specific classes. For this task, the following four classes are detected: Background, Slab, Beam, and Column.

The flowchart presented in Figure 9 visually summarizes the architecture and data flow for training or inference on images used for identification and segmentation of structural members like slabs and beams within a viaduct.

Figure 10 presents the process for the damage detection and segmentation task. Similarly to the structural task, it shows a deep learning pipeline employing DL models including U-Net and DeepLabV3+.

The input to the models is also shown as 320 × 160 × 3 data. This input again corresponds to images with three color channels: red (R), green (G), and blue (B).For this specific task, the output of the models is represented as 320 × 160 × 3 with three classes.The figure indicates that the segmentation aims to identify and differentiate types of damage and the background. For the damage detection task, the following three classes are identified: Background, Concrete spalling, and Reinforcement exposure.

The flowchart shown in Figure 10 represents the architecture and data flow for training or inference on images used for detection and segmentation of concrete damage and exposed reinforcement in the analyzed structural elements.

## 4. Comparison of the Results Obtained Using Both Architectures

Two variants of both the U-Net and DeepLabV3+ models were trained. In the first one, only RGB images were used with three channels as input. The second model additionally contained the depth channel, information on which was included in the data set. The learning process of the neural network spanned a maximum of 200 epochs, providing some callback on the evolution of the learning rate and avoiding early stopping of the learning process. In the first, the 2D model (RGB), categorical accuracy is equal to 92.08% while the loss is equal to 23.69%. In the three-dimensional model (RGB with depth channel), significantly better results were achieved. Categorical accuracy calculated on validation data was 97.22%, while the loss was equal to 8.39%. In practice, when inspection images are captured, they are not made in a stereoscopic version; hence, they will not have a depth channel. Therefore, the tests for 3D data should be treated as reference results to which the 2D model should aim.

### 4.1. Results for Task 1: Structural Component Identification

The dataset distribution for structural components of the analyzed viaduct (Figure 11) and non-structural elements shows that non-structural elements constitute the largest category at 58% (see Figure 3). Columns represent 18% of the dataset, while both the Slab and Beam components account for 12% each.
**Not Pretrained U-Net Model:** For the U-Net model without pretrained weights, the mIoU values for structural segmentation also fell within the range of 50% to 95%, using augmentations including Original, Brightness, Contrast, Gamma, Noise, Flip, Rotation, and All (Figure 12).**Not Pretrained DeepLabV3+ Model:** The version of DeepLabV3+ trained without initial weights from a large external dataset showed mIoU values in the range of 50% to 95% across the same set of augmentations (None, Brightness, Contrast, Gamma, Noise, Flip, Rotation, All) (Figure 13).**Pretrained DeepLabV3+ Model:** When subjected to various augmentations (None, Brightness, Contrast, Gamma, Noise, Flip, Rotation, All), this model achieved mIoU values ranging from 50% to 100% (Figure 14).

#### Differences Between Pretrained and Not Pretrained Models

Comparing the DeepLabV3+ models based on the use of pretrained weights reveals key differences:**Structural parts identification:** The pretrained DeepLabV3+ model demonstrated the potential to reach higher maximum mIoU values (up to 100%) (Figure 14) compared to the not pretrained DeepLabV3+ model (up to 95%) (Figure 13). This suggests that leveraging knowledge from a prior, potentially larger, dataset through pretraining can enhance the model’s capability to accurately segment structural elements, especially in achieving peak performance.**Concrete damage detection:** Both the pretrained DeepLabV3+ model (range 0–35%) (Figure 15) and the not pretrained DeepLabV3+ model (range 0–35%) (Figure 16) exhibited a similar range of mIoU values. The relatively low mIoU values observed for damage detection across all models (maximum 35% for DeepLabV3+ (Figure 15 and Figure 16) and 7% for U-Net (Figure 17)) indicate that identifying damage is a considerably more challenging task than segmenting the main structural components. In this specific task, pretraining the DeepLabV3+ model did not appear to offer a distinct advantage over training from scratch, based on the range of results presented.

Furthermore, comparing the models, DeepLabV3+ (both pretrained and not pretrained) achieved substantially better results in damage detection (range 0–35%, Figure 15 and Figure 16) than the not pretrained U-Net (range 0–7%, Figure 17). For structural segmentation, the not pretrained U-Net (range 50–95%, Figure 12) performed comparably to the not pretrained DeepLabV3+ (range 50–95%, Figure 13), while the pretrained DeepLabV3+ showed the potential for slightly higher maximum accuracy (up to 100%, Figure 14).

In summary, the results indicate that structural parts segmentation is a task where high mIoU values are achievable, and pretraining DeepLabV3+ potentially improves maximum accuracy. Concrete damage detection, conversely, is significantly more difficult, yielding much lower mIoU values, and pretraining DeepLabV3+ did not show a clear advantage over the not pretrained version within the observed ranges. Different data augmentation strategies were applied and influenced the reported metrics for all evaluated models in both tasks.

### 4.2. Results for Task 2: Structural Damage Detection

In the structural damage detection task (Figure 10), which includes cracks and exposed reinforcement, models were evaluated under a range of image augmentations: None, Brightness, Contrast, Gamma, Noise, Flip, Rotation, and All.

**Pretrained DeepLabV3+ Model:** When trained with standard categorical cross-entropy loss, the pretrained DeepLabV3+ model produced low mIoU values for damage classes, ranging-from 0% to 35%, depending on the augmentation used (see Figure 18). Performance on the reinforcement class was especially poor, with many predictions missing entirely.**Trained-from-scratch DeepLabV3+ Model:** Similarly, the DeepLabV3+ model without pretrained weights exhibited comparable mIoU values (0–35%), with minimal improvements under specific augmentations (Figure 16).**Trained-from-scratch U-Net Model (Baseline):** Initially, the U-Net model trained from scratch using categorical cross-entropy achieved the lowest mIoU values among all tested architectures—typically in the range of 0% to 7% across augmentations (Figure 17).**Improved U-Net Model with Weighted Focal Tversky Loss function:** After implementing a weighted focal Tversky loss function with recall-favoring hyperparameters (α=3, β=7, γ=1.5) and class weights [05,3,65] to handle severe class imbalance, the U-Net model achieved a substantial performance boost (Table 2 and Table 3). The best configuration reached an IoU of 48 for cracks and 44 for reinforcement, with corresponding F1 scores of 65 and 61, respectively. This demonstrates the importance of loss adaptation for fine damage segmentation. Background segmentation accuracy remained high (IoU = 98), confirming that foreground detection was improved without sacrificing overall stability.**Improved DeepLabV3+ Model with Weighted Tversky Loss function:** To evaluate the impact of loss function choice on damage segmentation performance, we compared the commonly used **categorical cross-entropy (CCE)** loss with a **weighted focal Tversky loss** formulation. The results of both configurations were computed on the same test set using the same U-Net architecture and are presented in Table 4.The results indicate that both loss functions yield similarly high performance for the background class (IoU ≈ 0.97–0.98). However, substantial differences were observed for the damage classes:-For **cracks**, the weighted Tversky loss improved the IoU from 0.20 to 0.42 and increased the F1 score from 0.39 to 0.59. This reflects a better balance between precision and recall, which is especially important for detecting small and thin regions.-For **reinforcement**, the Tversky-based configuration significantly outperformed CCE in all metrics, improving IoU from 0.12 to 0.38 and F1 from 0.29 to 0.55.These improvements are attributed to the Tversky loss’s ability to penalize false negatives more heavily, which aligns with the safety-critical nature of structural damage detection, where missing a damaged area is more critical than false alarms. Furthermore, class imbalance was explicitly addressed through weighting, allowing the network to better learn underrepresented classes. The weighted Tversky loss demonstrates a clear advantage over categorical cross-entropy for fine-grained segmentation tasks involving small and imbalanced damage regions and was therefore adopted in our final model configuration. A visual comparison between U-Net results and DeepLabV3+ with the application of Tversky loss function can be found in Figure 19.

### 4.3. Train and Validation Evaluation for the U-Net Model

To assess model generalization across dataset splits, we present a detailed breakdown of performance metrics on the training, validation, and test sets for U-Net models (as most efficient on our real photo), trained using both the categorical cross-entropy (CCE) and weighted Tversky loss functions. Results are reported per class in Table 5, with all values expressed as percentages.

The results confirm that models trained with the weighted Tversky loss achieve significantly better segmentation of the minority damage classes, particularly cracks and reinforcement. For example, in the test set, the IoU for cracks improved from 32.14% (CCE) to 47.87% and for reinforcement from 11.07% to 43.87%. This improvement is consistent across training and validation splits, reflecting better generalization and damage sensitivity.

Furthermore, a comparison of metrics across dataset splits offers insight into model behavior:**Background class** performance is highly stable for both loss functions (IoU consistently ≈ 98%), showing the model’s ability to accurately segment dominant classes without overfitting.**Cracks**: For categorical cross-entropy, there is a notable gap between train (39.4%) and test (32.1%) IoU, suggesting mild overfitting. In contrast, the Tversky loss achieves higher and more balanced performance across all splits, even slightly outperforming train IoU on test data (47.9% test vs. 35.9% train), indicating robust generalization to unseen samples.**Reinforcement**: Under categorical cross-entropy, segmentation nearly collapses across all splits (test IoU: 11.1%, train: 8.6%). This class suffers from extreme imbalance and sparsity. The Tversky loss, in contrast, leads to a significant increase in reinforcement segmentation (IoU: 43.9% test, 48.2% val), highlighting the effectiveness of weighting and false-negative sensitivity in loss formulation.

Together, these observations underline the importance of tailored loss functions when addressing fine-grained and imbalanced damage detection tasks. The weighted Tversky loss not only boosts absolute performance but also stabilizes training dynamics across dataset splits, reinforcing its suitability for practical deployment scenarios.

The Tversky-based configuration reduces false negatives—a critical benefit in safety-relevant applications where missed damage is more consequential than false alarms. Conversely, CCE performs well for background segmentation but fails to adequately capture sparse damage classes. These findings reinforce the suitability of the weighted Tversky loss for highly imbalanced, fine-grained damage segmentation tasks.

### 4.4. Real-World Evaluation on Annotated Viaduct Images

To assess model generalization beyond synthetic data, we performed an additional evaluation on the manually labeled Second using the MATLAB 2024a labeler, a set of 16 real-world viaduct images. Four classes were annotated: slab, beam, column, and non-structural background. Figure 20 shows a sample qualitative comparison of our model’s predictions with the reference annotations presented in the paper [28]. Table 6 reports the quantitative performance, including class accuracy, precision, recall, F1 score, and IoU.

These results indicate that despite being trained solely on synthetic data, the model retains a moderate degree of generalization to real-world imagery, especially for well-defined classes such as columns. Further improvements are expected with the integration of real-world samples in fine-tuning or domain adaptation steps.

## 5. Conclusions

In conclusion, the imperative role of maintaining traffic infrastructure in an optimal technical condition for ensuring road and rail traffic safety cannot be overstated. As the traffic infrastructure expands, the challenges associated with its maintenance grow exponentially, necessitating a robust system for continuous monitoring. The integration of vision techniques with artificial intelligence has emerged as a powerful solution for the efficient automation of inspection processes.

The deployment of various monitoring tools, including installed cameras, remotely controlled vehicles, and autonomous drones, enables the independent capture of images, providing a comprehensive view of the infrastructure. The subsequent utilization of artificial intelligence, particularly deep learning, in image processing facilitates the identification of potential damage within the structures. Training deep neural networks is essential for endowing these systems with the capability to recognize and assess damage accurately.

Based on the results presented in Section 4, we can conclude that it is possible to apply the neural architectures analyzed in this study to computer-vision-based structural health monitoring (CV-SHM). However, looking at the investigated *Mean IoU* metric, we can state that DeepLabV3+ outperforms U-net in both segmentation tasks. DeeplabV3+ achieves *Mean IoU* equal to 87% in the structural component identification task and 35% in the structural damage detection. U-net for the same tasks reached 76% and 13%, respectively.

While our study focuses on optimizing the loss function and training dynamics, we acknowledge that architectural adaptations—such as attention modules, multi-scale fusion, or boundary-aware refinement—may further enhance the segmentation of small-scale features. These techniques represent a valuable direction for future research, particularly in conjunction with the proposed loss-based improvements. Our future goal will also be related to improving the performance of the model to be able to use it in the semantic segmentation of photographs of real civil engineering structures. Additionally, it would be interesting to further investigate the possibilities of using other types of neural network architectures for CV-SHM applications, in particular, the vision transformer mentioned in the introduction.

## Figures and Tables

**Figure 1 sensors-25-04698-f001:**
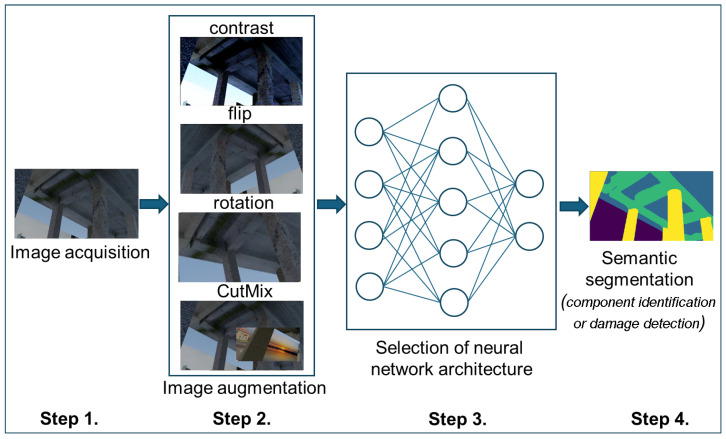
Flowchart of the training process.

**Figure 2 sensors-25-04698-f002:**
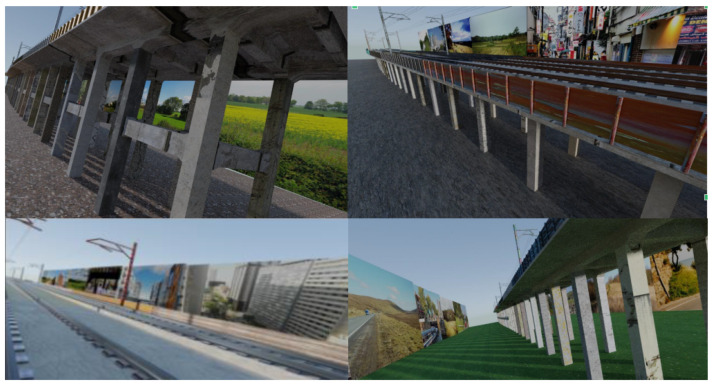
Sample images of railway viaducts included in the Tokaido synthetic dataset [34].

**Figure 3 sensors-25-04698-f003:**
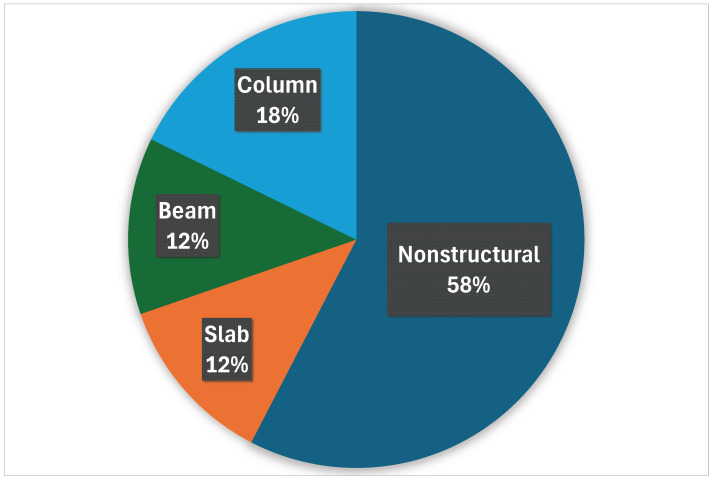
Pixel distribution of individual classes in particular segmentation masks.

**Figure 4 sensors-25-04698-f004:**
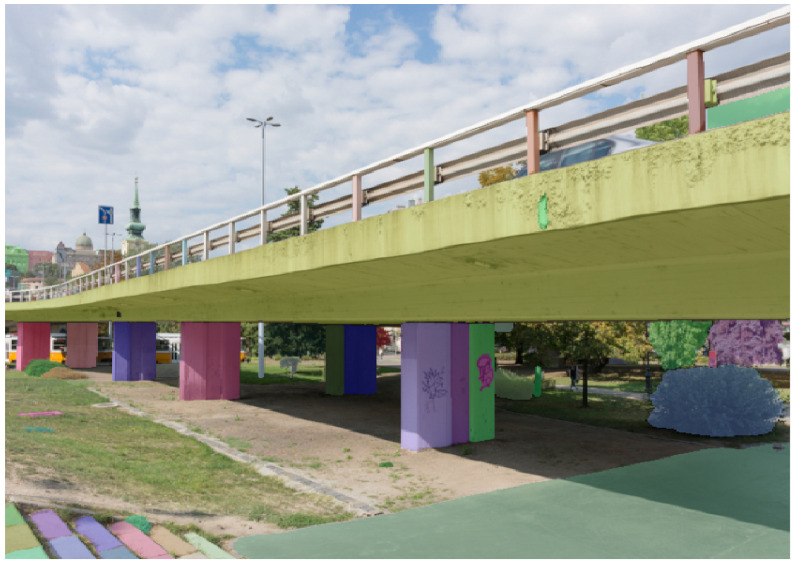
Instance segmentation using *Segment Anything* [33] (on the photograph: viaduct located in Budapest).

**Figure 5 sensors-25-04698-f005:**
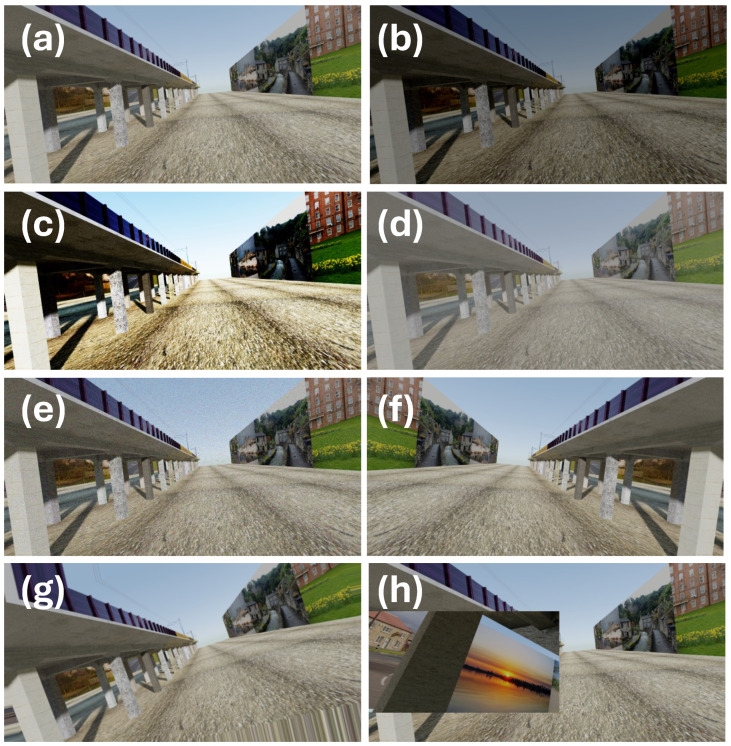
Various approaches to image augmentation applied in the training process: (**a**) ground truth, (**b**) brightness adjustment, (**c**) contrast modification, (**d**) gamma correction, (**e**) noise addition, (**f**) horizontal flip, (**g**) arbitrary rotation and (**h**) CutMix operation.

**Figure 6 sensors-25-04698-f006:**
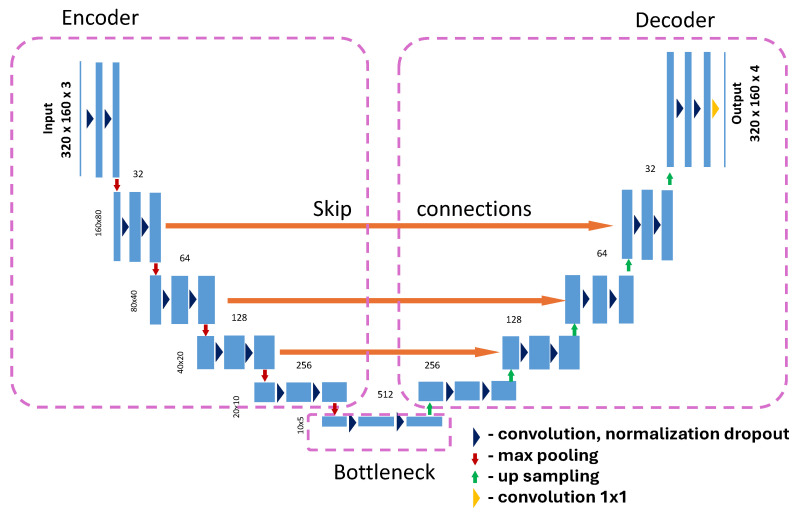
U-Net architecture according to the paper by Ronneberger [9].

**Figure 7 sensors-25-04698-f007:**
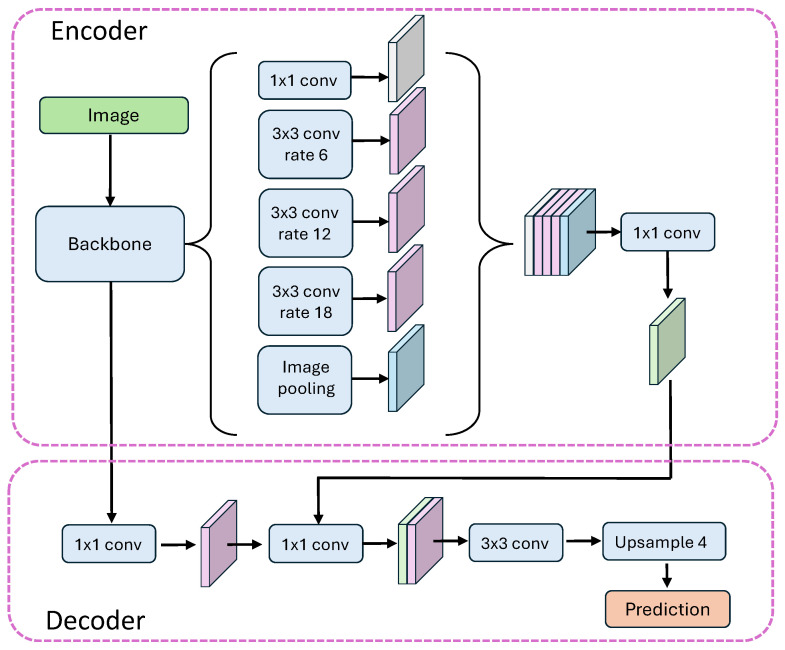
DeeplabV3+ architecture according to the paper by Chen et al. [35].

**Figure 8 sensors-25-04698-f008:**
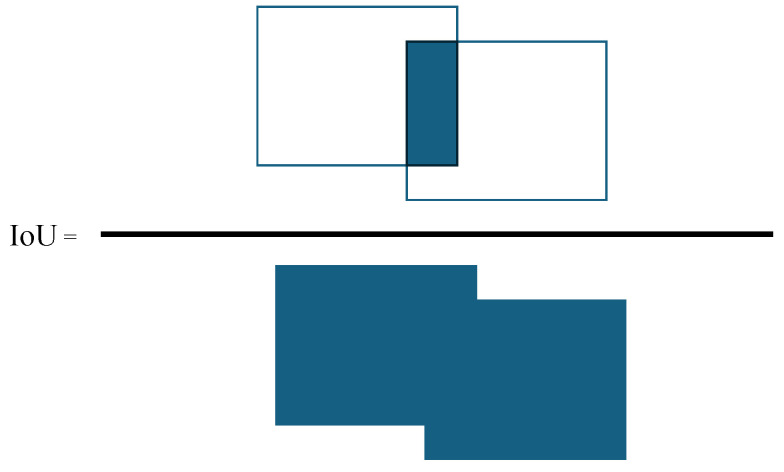
IoU = Area of Overlap/Area of Union.

**Figure 9 sensors-25-04698-f009:**
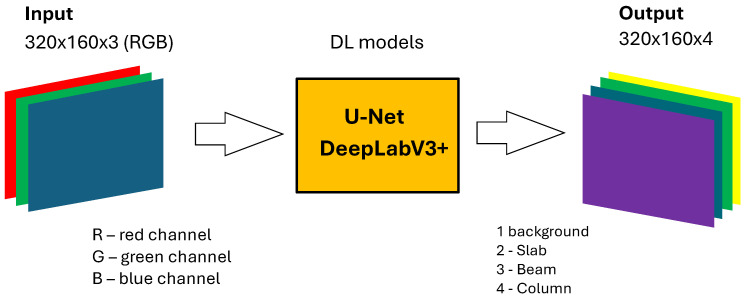
Input and output layers for the structural component identification task.

**Figure 10 sensors-25-04698-f010:**
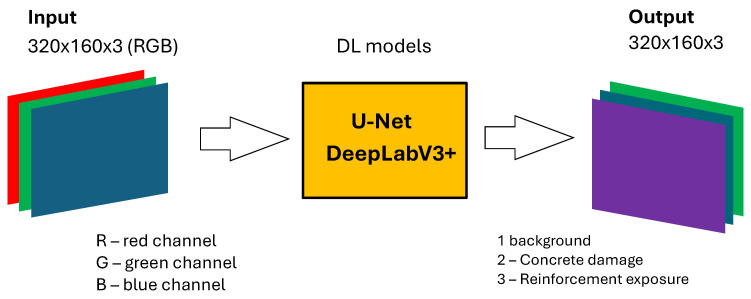
Input and output layers for damage detection task.

**Figure 11 sensors-25-04698-f011:**
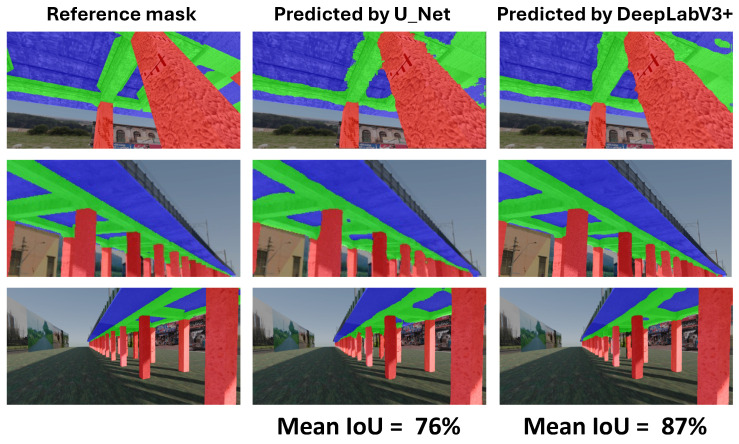
Comparison of the results of component identification obtained using both U-Net and DeepLabV3+ architectures.

**Figure 12 sensors-25-04698-f012:**
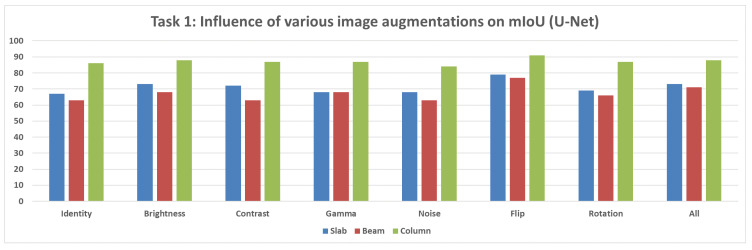
Mean IoU metric comparison of various augmentations applied for the U-Net model.

**Figure 13 sensors-25-04698-f013:**
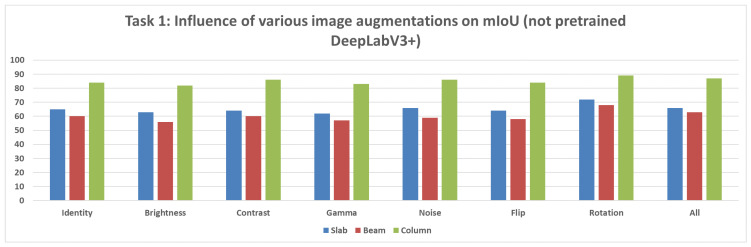
Mean IoU metric comparison of various augmentations applied for the not pretrained DeepLabV3+ model.

**Figure 14 sensors-25-04698-f014:**
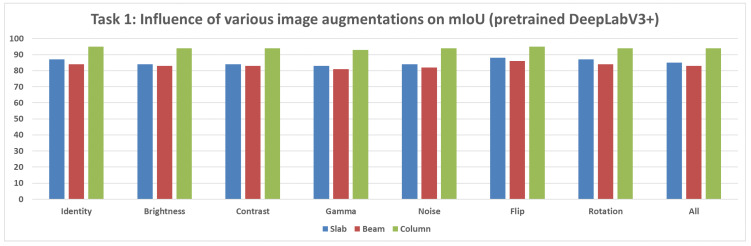
Mean IoU metric comparison of various augmentations applied for the pretrained DeepLabV3+ model.

**Figure 15 sensors-25-04698-f015:**
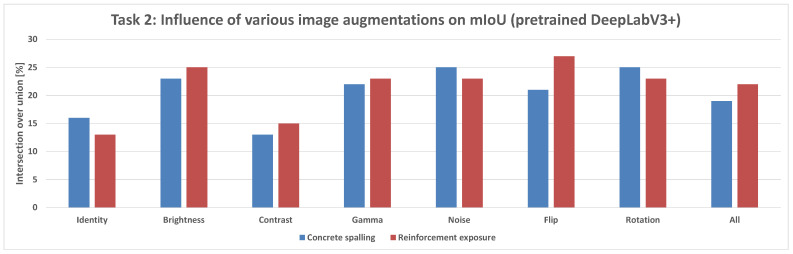
Mean IoU metric comparison of various augmentations applied in the pretrained DeepLabV3+ model for the damage prediction task.

**Figure 16 sensors-25-04698-f016:**
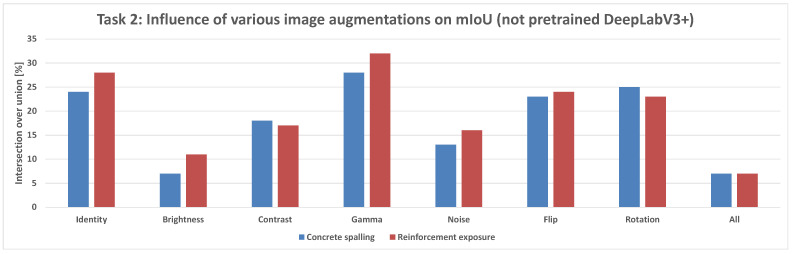
Mean IoU metric comparison of various augmentations applied for the not pretrained DeepLabV3+ model in the damage prediction task.

**Figure 17 sensors-25-04698-f017:**
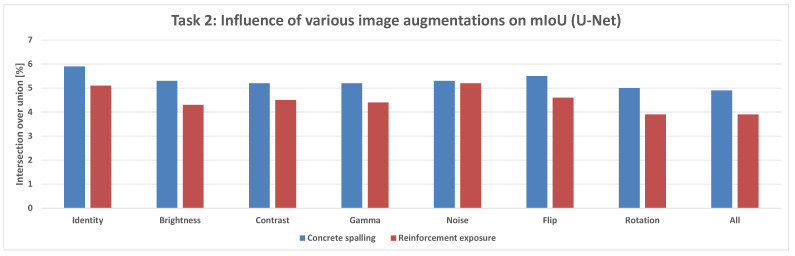
Mean IoU metric comparison of various augmentations applied for the U-Net model in the damage prediction task.

**Figure 18 sensors-25-04698-f018:**
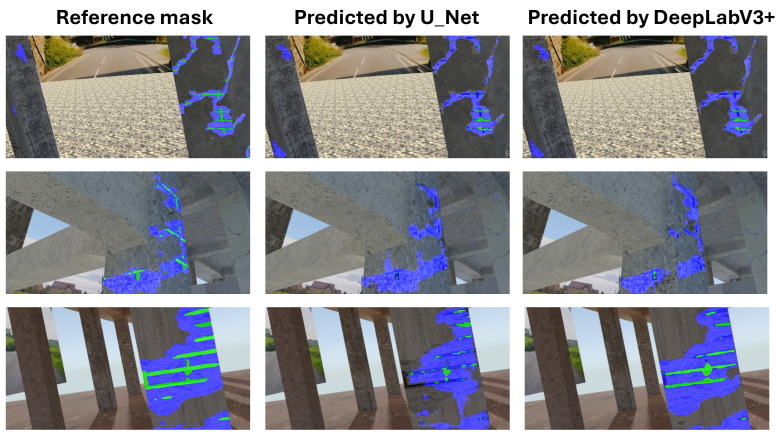
Comparison of the results of damage detection obtained using both U-Net and DeepLabV3+ architectures with a categorical cross-entropy loss function.

**Figure 19 sensors-25-04698-f019:**
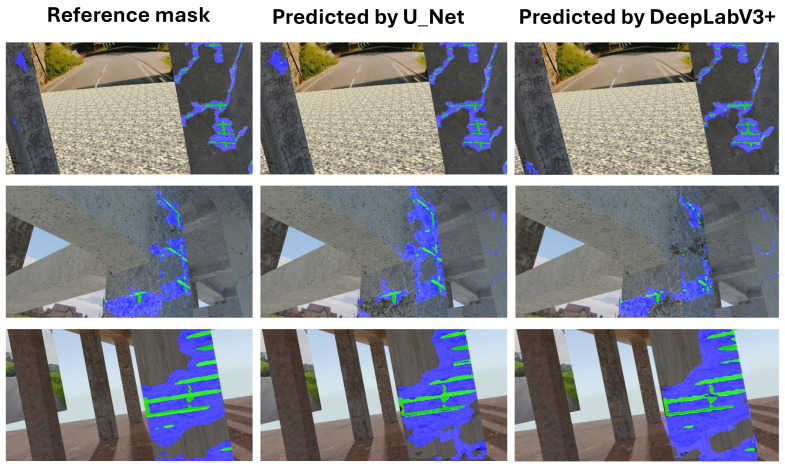
Comparison of the results of damage detection obtained using both the U-Net and DeepLabV3+ architectures with a weighted Tversky loss function.

**Figure 20 sensors-25-04698-f020:**
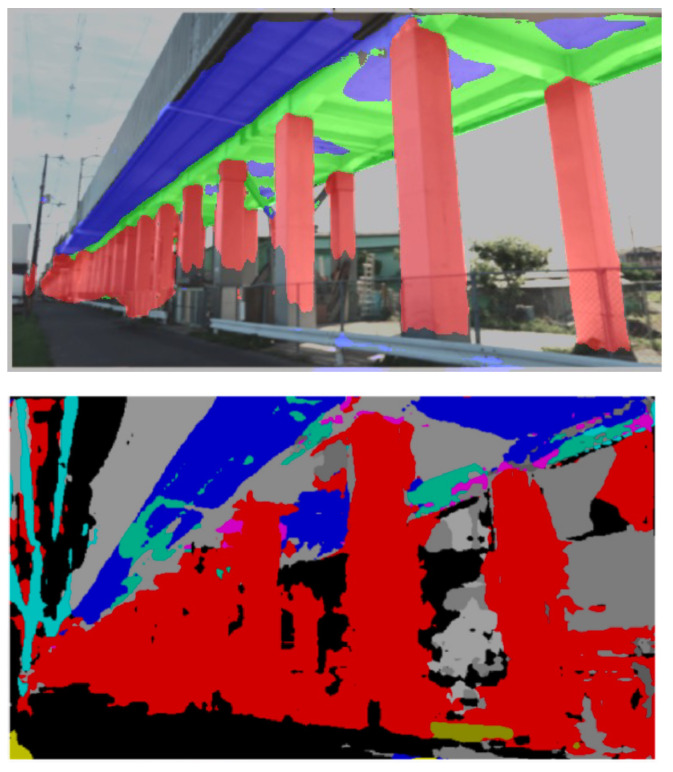
Evaluation of the structural segmentation model with real photos (upper photo). Lower segmentation image, for comparison, taken from paper [28].

**Table 2 sensors-25-04698-t002:** Per-class segmentation performance for U-Net.

Class	Class Acc. [%]	Precision [%]	Recall [%]	F1 Score [%]	IoU [%]
Background	98	98	99	99	98
Cracks	98	53	45	61	32
Reinforcement	100	19	21	0	11

**Table 3 sensors-25-04698-t003:** Final segmentation performance using the best U-Net configuration with weighted focal Tversky loss. Metrics are rounded to two decimal places.

Class	Class Acc. [%]	Precision [%]	Recall [%]	F1 Score [%]	IoU [%]
Background	98	99	99	99	98
Cracks	99	61	69	65	48
Reinforcement	00	79	49	61	44

**Table 4 sensors-25-04698-t004:** Comparison of segmentation performance using categorical cross-entropy and weighted Tversky loss.

Loss	Class	Class Acc.	Precision	Recall	F1 Score	IoU
CCE [%]	Background	97	98	99	99	97
	Cracks	98	0.37	31	39	20
	Reinforcement	99	0.19	25	29	12
Tversky [%]	Background	98	99	99	99	98
	Cracks	98	57	61	59	42
	Reinforcement	100	67	47	55	38

**Table 5 sensors-25-04698-t005:** Train and validation performance for categorical cross-entropy (CCE) and weighted Tversky loss for U-Net damage model. All values are in %.

Loss	Split	Class	IoU	F1 Score	Precision	Recall
CCE	Train	Background	98.11	99.28	98.68	99.41
		Cracks	39.42	71.07	59.38	53.97
		Reinforcement	8.55	0.00	12.85	20.32
	Validation	Background	98.04	99.28	98.66	99.36
		Cracks	38.65	69.54	58.92	52.90
		Reinforcement	9.86	0.00	15.41	21.50
Tversky	Train	Background	98.15	99.07	98.99	99.15
		Cracks	35.87	53.02	45.69	62.55
		Reinforcement	24.31	39.65	41.33	37.12
	Validation	Background	98.25	99.12	99.04	99.20
		Cracks	49.16	65.93	59.16	74.41
		Reinforcement	48.23	65.08	75.27	57.32

**Table 6 sensors-25-04698-t006:** Real-world segmentation results on 16 annotated viaduct images.

Class	Class Acc. [%]	Precision [%]	Recall [%]	F1 Score [%]	IoU [%]
Non-structural	50	40	52	45	29
Slab	51	39	26	31	19
Beam	83	08	14	10	05
Column	88	37	27	32	19

## Data Availability

The dataset used in this study is publicly available on the Zhejiang University academic website at https://person.zju.edu.cn/en/H121007#964615.

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
