# Peer review of "Structural Component Identification and Damage Localization of Civil Infrastructure Using Semantic Segmentation"

_sensors, 2025, doi:10.3390/s25154698_

Round 1

Reviewer 1 Report

Comments and Suggestions for Authors

Seeing attachment.

Reviewer 2 Report

Comments and Suggestions for Authors

The paper is devoted to a problem of defect detection by the means of computer vision. The authors demonstrate a possibility to detect and localize damages in road an building infrastructure with the help of neural-network based image segmentation. The authors also address the problem of insufficient training data and propose a solution based on augmentation of synthetic data. It is shown that this way gives better resulting performance. The paper contains all necessary parts, the language and logic are clear. There are the following drawbacks.

1. Proposed Cutmix technique is very rough. As one can see, it is simply pasting a rectangular fragment of some other image, that forms a sharp regular vertical and horizontal borders. It was discussed recently that in this case neural network is likely to learn to detect this rectangle and ignore its contents. Real data do not contain extraneous rectangles. So the existing Cutmix does not augment much. More elaborate version of introducing textures is required.

2. Description of augmentation methods is incomplete. Formula in line 233 requires 'clip' operation for brightness value. Formula in line 240 requires 'clip' operation for coordinates. Cutmix operation is not described formally, at least its parameters (position and size of pasted fragment etc) should be given in table 1.

Reviewer 3 Report

Comments and Suggestions for Authors

This paper investigates the application of convolutional neural networks (U-Net and DeepLabV3+) for semantic segmentation of civil infrastructure, focusing on structural component identification and damage localization. The models are trained exclusively on a synthetic dataset (Tokaido) and evaluated on both synthetic and real-world images. The authors demonstrate promising results, particularly highlighting the superior performance of DeepLabV3+ over U-Net.

Major Weaknesses:

- Lack of a Proposed or Novel Architecture: The paper presents results using two conventional architectures (U-Net and DeepLabV3+), both of which are several years old. There is no proposed method or architectural innovation. As such, the technical novelty of the paper is minimal. The use of only legacy baselines without comparison to current state-of-the-art models—such as Vision Transformers or recent foundation models for segmentation (e.g., Segment Anything, Mask2Former, SegFormer, etc.)—is a significant limitation.

- No Proper Baseline Justification or Ablation Study: The paper fails to justify why U-Net and DeepLabV3+ were chosen as the only methods for comparison. Additionally, no ablation studies or sensitivity analyses are presented to understand the impact of key factors (e.g., augmentation, input resolution, backbone choice) on performance.

- Evaluation on Real-World Images Is Anecdotal: While the paper claims promising results on real-world imagery, there is no formal test set, metric-based evaluation, or ground-truth annotation presented for these images. Consequently, any claims about generalization from synthetic to real-world data remain speculative.

- Figures Lack Clarity and Professionalism: Several figures (e.g., Figures 14 and 15) are visually blurry or poorly rendered, making interpretation difficult. Figure 8 is a direct copy from an online source and does not appear to be cited properly. The overall visual quality of the illustrations undermines the scientific presentation of the results.

- Dataset Split and Experimental Design Are Unclear: It is not clear how the dataset was split between training, validation, and test sets. It appears that only a synthetic dataset is used for training and validation, and that validation metrics are being reported without a clear test set. The distinction between validation and test performance must be made explicit, especially when real-world generalization is a core claim.

- Lack of Standard Evaluation Protocols: The experimental results lack standard statistical reporting. No confidence intervals or standard deviations are provided, and it's unclear how many trials were conducted for each experiment. This omission reduces the reliability and reproducibility of the findings.

Required Revisions:

1. Compare with More Modern Baselines:

   - Include transformer-based segmentation models or recent lightweight architectures designed for embedded deployment (e.g., SegFormer, Fast-SCNN).
   - Justify why only classical CNN-based models were selected when the field has moved well beyond them.

2. Clarify Dataset and Evaluation Protocols:

   - Provide detailed information on how the dataset was split into training, validation, and test sets.
   - Clearly indicate whether real-world performance was quantitatively assessed, and if not, explain why.

3. Improve Figures and Visual Clarity:

   - Replace blurry figures with high-resolution plots.
   - Avoid reusing figures from online sources unless properly cited. If using externally created graphics (e.g., Figure 8), provide appropriate credit and permissions.

4. Provide Statistical Analysis:

   - Report the number of training runs and include standard deviations or confidence intervals for all metrics.
   - Include plots that show learning stability over time, and indicate if overfitting was observed.

5. Explain Figure 11 More Clearly:

   - The learning curve in Figure 11 lacks context. What exactly is being shown? What does the curve imply about the model’s convergence or generalization?

6. Clarify Contribution and Novelty:

   - As currently framed, the paper reads more like an engineering application note rather than a research contribution. Make clear what the novelty is—if it's in synthetic dataset generation, domain transfer, or deployment, this needs to be explicitly stated and validated.

Questions to Address in the Rebuttal:

- Why were U-Net and DeepLabV3+ selected as the only models? Are the results still competitive against more modern transformer-based architectures?
- How were the datasets split between training, validation, and testing? Was a separate test set used?
- What specific contribution does the paper make beyond applying existing models to a new dataset?
- Can the authors provide proper citations and permissions for all reused visuals (e.g., Figure 8)?
- Are the models trained with sufficient regularization to avoid overfitting on synthetic data?

Round 2

Reviewer 1 Report

Comments and Suggestions for Authors

This version is better, and the issues have been well addressed.

Author Response

Comment 1: This version is better, and the issues have been well addressed.

Response 1: The Authors would like to thank the Reviewer for accepting our paper in its current form.

Reviewer 2 Report

Comments and Suggestions for Authors

The authors have responded the remarks and provided sufficient explanation in the text. The paper can be published now.

Author Response

Comment 1: The authors have responded the remarks and provided sufficient explanation in the text. The paper can be published now.

Response 1: The Authors would like to thank the Reviewer for accepting our paper in its current form.

Reviewer 3 Report

Comments and Suggestions for Authors

- Also, It would be useful to include a breakdown of the performance for each labeling function and label model, even if it's just on a subset of the validation or test dataset, rather than only reporting the final classification score.

I have no major concerns regarding the broader impact of this work. In my opinion, it is suitable for acceptance and publication, pending the editorial review.
